# Caught on camera: Field imagery reveals the unexpected importance of vertebrates for biological control of the banana weevil (*Cosmopolites sordidus* Col. Curculionidae)

**Paul Tresson**[1,2,3,4¤a¤b]*, **Philippe Tixier**[1,2], **William Puech**[4], **Bernard Abufera**[2,3], **Antoine Wyvekens**[2,3], **Dominique Carval**[1,2,3]

**1** CIRAD, UPR GECO, Montpellier, France, **2** GECO, Univ Montpellier, CIRAD, Montpellier, France, **3** CIRAD, UPR GECO, Saint-Pierre, France, **4** LIRMM, Université de Montpellier, CNRS, Montpellier, France

¤a Current address: CIRAD, UMR AMAP Montpellier, France
¤b Current address: Valorhiz, Montpellier, France
* paul.tresson@cirad.fr

**Data Availability Statement:** The image dataset is publicly available at dataverse.cirad.fr repository at the following DOI: https://doi.org/10.18167/DVN1/5F7FVI.

## Abstract

Understanding of ecological interactions is necessary for the application of biological control. Banana is the second most produced fruit worldwide and the banana weevil (*Cosmopolites sordidus*) is the most important pest of banana and plantain. Its biological control remains challenging because of the robustness and cryptic behaviour of the adult and the hidden development of larval stages. Researchers therefore tend to favour conservation biological control of this pest. The commonly used methods for measuring the effects of natural enemies on the regulation of this pest focus on invertebrates and may underestimate the role of vertebrates on biological control. Using cameras, we recorded the predation of sentinel adult weevils in banana plots in La Réunion island that differed in weevil infestation levels and in animal biodiversity. To facilitate image analysis, we used background subtraction to isolate moving parts of image sequences and thus detect predators and predation events. Our cameras recorded only vertebrates as predators of adult banana weevils. The most important predator appeared to be the Asian shrew (*Suncus murinus*), which was responsible for 67% of the predation events. Other predators included the house mouse (*Mus musculus*), the oriental garden lizard (*Calotes versicolor*), and the guttural toad (*Sclerophrys gutturalis*). The exact time of predation events were determined from the images metadata. It was thus possible to identify predator foraging periods that coincided with activity of adult weevils. Our results confirm that images provide useful information for biological and ecological studies. Along with other recent studies, our results suggest that the role of vertebrates in biological control may be underestimated. Based on these results, we advocate for several management implications such as the installation of hedges, grasslands, and ponds to favour these vertebrate predators of the banana weevil, possibly also favouring other vertebrate and invertebrate natural enemies.

**Funding:** This work was carried out as part of the CIRAD DPP COSAQ agronomical research programme (activities 2015–2021) funded by a grant from the European Community (ERDF) and the Conseil Régional de La Réunion. This work was also supported by the French National Research Agency under the Investments for the Future Program, referred to as ANR-16-CONV-0004. The Ph.D. thesis of P. Tresson was funded by CIRAD and #DigitAg grants The funders had no role in study design, data collection and analysis, decision to publish, or preparation of the manuscript.

**Competing interests:** The authors have declared that no competing interests exist.

## 1. Introduction

Sustainable agriculture relies on ecosystem functions [1], including the control of pests and diseases. A necessary step for the successful control of pests is the accurate assessment of pest regulation and the identification of species responsible for such regulation. This is especially true for conservation biological control as it relies on habitat management to favor occurrence and functionality of natural enemies present on site. Then, key species or assemblage of natural enemies and interactions between them and the pest may not be known [2–4]. to better understand the biological control service it is needed to identify the species at play and their interactions. Data on pest regulation by natural enemies may be obtained by several existing methods such as correlative studies on pest-predator abundances (*e.g.*, [5]), molecular and immunological gut content analysis (*e.g.*, [6, 7]), predator exclusion experiments (*e.g.*, [8]), or sentinel-prey experiments (*e.g.*, [9, 10]). However, as informative as they are, these existing methods to sample biodiversity and study interactions between species may have biases or limitations.

Stable isotopes, for example, have widely been used to analyse the position of species in trophic networks. This method provides information on the structure and modifications of the trophic network but is not sufficiently precise to prove the trophic relationship between two species [11, 12]. Another possibility is the use of metabarcoding or analysis of gut contents [13, 14], but these methods may fail to provide important information on prey such as the prey development stage, hyper-predation, failed predation, or scavenging [7, 15].

Although sentinel prey experiments are informative in estimating pest regulation, long exposure times may lead to biases such as the attraction of opportunist predators and the alteration of foraging behaviour [9]. Without monitoring or use of immunomarking techniques [7] it may be difficult to identify the predator responsible for the disappearance of the prey at the end of the experiment. Indeed, in 40 of the 57 studies reviewed by Lövei and Ferrante [9], the predator responsible for the consumption of the sentinel prey remained unidentified. There is thus a great potential in coupling sentinel prey approaches with monitoring methods in order to identify predators.

All these methods may also involve sampling biases. For instance, pitfall trapping, which is commonly used to sample litter arthropods, has been shown to bebiased [16–18]. One recent study using time-lapse cameras revealed that cockroaches and ants were able to escape pitfall traps [19]. These species would therefore be underestimated in biodiversity samplings.

To counter these limitations and biases, researchers are increasingly relying on images as sources of information [20]. Manual image analysis, however, remains time consuming (*e.g.*, [21]). One solution to this problem is to rely on automated image analysis with, for instance, machine-learning techniques. To date, machine learning in ecology has mostly been used for automatic identification and classification of species [22–24]. More recent uses include for example species detection and tracking [25] or interaction analysis [26]. The use of automated image analysis can allow a detailed study of the recruiting dynamics of ants or of the number of individuals needed to seize a prey animal, enabling the reconstruction of the observed interaction network [26]. In some cases, however, image analysis can be performed with less demanding methods, such as background subtraction or color thresholding, and yield sufficient results without the need of a specific labelled dataset.

Recent experiments with camera-monitored sentinel prey have revealed unexpected importance of vertebrates for biological control of pests (brown planthoppers, [27, 28]; mealworms, [29]). As standard methods used to sample natural enemies are often designed to sample arthropods, the role of vertebrates could easily be underestimated. Vertebrate insectivores, however, may have a large impact on ecosystems as predators of pests [30, 31] or possibly as intra-guild predators [8, 31]. To date, birds (*e.g.*, [32–34]) and bats (*e.g.*, [35, 36]) are the most

studied vertebrates for biological control of insect pests. However, amphibians (*e.g.*, [27, 37]), lizards (*e.g.*, [38]), and rodents (*e.g.*, [29]) may also contribute to pest regulation.

In this study, we focused on the banana weevil *Cosmopolites sordidus* (Col. Curculionidae), which is the major pest of banana and plantain around the world [39]. Females lay eggs at the bottom of the banana's pseudo-stem or in the corm, and larvae later develop in the corm (the bulb of the banana plant). The resulting internal damage weakens the plant and can lead to yield losses and crop failure in newly planted stands [39]. The banana weevil has a nocturnal and cryptic behaviour, moving at night and on the ground [40]. The hidden larval development of *C. sordidus* hinders the predation of these stages. Furthermore, the thick cuticle and physical robustness of the adult of *C. sordidus* leave biological control methods based on invertebrate predators difficult for this stage. [41]. To date, the most investigated predators of the banana weevil are arthropods, especially ants [42], beetles [43, 44], and earwigs [6]. All these studies focused on the regulation of eggs and larval stages. Because of this arthropod bias, the role of vertebrates in the regulation of *C. sordidus* is considered anecdotal [41].

In this paper, we used cameras to monitor sentinel prey (alive *C. sordidus* adults) to identify possible predators, allowing a more in-depth assessment and understanding of the regulation of the banana weevil. We conducted these observations in five banana plots in La Réunion island, where banana cultivation involves diverse cultural practices and weevil infestation levels. We attempted to answer the following questions: (1) Which predators attack *C. sordidus*? (2) When during the 24-hour period does predation occur? (3) Which predators are most likely to regulate *C. sordidus*?

## 2. Materials and methods

### 2.1. Location and studied plots

The experiment was conducted in dessert banana (*Musa*, *spp*. AAA Cavendish group) plantations on La Réunion Island (French overseas territory). Although five plots were designated in a small area to minimize differences in soil and climate, the plots were selected to include diverse cultural practices and *C. sordidus* levels (Table 1). The plots ranged in size from 0.38 to 2.7 ha and the farmers did not change their cultural practices before or during the experiment. All plots, labelled hereafter BM, LP, PC, PE, and SL, were located between 21˚15'35"S and 21˚ 18'36"S, and 55˚24'52"E and 55˚30'05"E. Altitudes ranged from 9 to 223 m above sea level. Distances between the plots ranged from 0.5 km to 11.4 km. As weevils disperse slowly and are usually aggregated in the field [40], all the plots were considered as statistically independent. During the experiment, temperatures ranged between 18˚C and 31˚C, and precipitation was relatively low (< 150 mm of rain over 3 months).

To assess the *C. sordidus* infestation levels at the beginning of the experiment, we deployed weevil traps with pheromones (Cosmolure®, Chemtica, Costa Rica) two months before the experiment began. For each plot, the trapping was carried out with two traps during four

**Table 1. Background information for the five plots.** Age indicates the number of years that bananas were continuously grown in the plot. Captures indicates the number of weevils captured in the plot in the 2-month period before the study began. Under Cultural practices, hedges are mostly represented by sugarcane.

| Code | Location | Age (years) | Area (ha) | Captures | Cultural practices |
|------|----------|-------------|-----------|----------|--------------------|
| BM | Bassin-Martin | 1 | 0.38 | 14 | Grass cover, associated culture (papaya) |
| LP | Ligne-Paradis | 6 | 1.62 | 2 | Grass cover |
| PC | Pierrefonds | 1 | 2.7 | 0 | Grass cover, hedges, ponds |
| PE | Pierrefonds | 9 | 0.86 | 0 | Bare soil, hedges, ponds |
| SL | Saint-Louis | 7 | 2.4 | 5 | Bare soil, hedges |

weeks (two weeks in August and two weeks in September). As shown on Table 1, the infestation level was highest in the BM plot, was intermediate in LP and SL plots, and was lowest in PE and PC plots.

## 2.2. Sentinel prey tiles

At the start of the experiment, five sentinel prey tiles were randomly deployed in each of the five plots, leading to 25 different tiles. Between the start of the experiment on September 15th 2020 and its end on December 1st 2020, five recording sessions were conducted for each sentinel tile, resulting in a total of 125 recordings. Although all tiles within a plot were recorded within 24 hours, records were taken on different days for the five plots with at least 5 days between sequential sessions in a plot.

Sentinel *C. sordidus* were collected in plots located in the same region as our studied plots. Then they were reared in controlled conditions during four to eight weeks before the experiment. Each sentinel $30 \times 30$ cm prey tile was placed on the soil surface, and included two healthy *C. sordidus* adults. Tiles were made of ceramic, with a light grey and rough surface. The adults were attached to the tile with a 10-cm-long piece of nylon fishing line, which was attached to the top of the cephalothorax of each adult with a drop of cyanoacrylate glue. The other end of the fishing line was tied to a second piece of fishing line that was tightened across the tile (Fig 1a). Sections of banana pseudostems were placed on the tile to provide potential cover for the weevils. Our preliminary observations indicated that, unless attacked by a predator, the sentinel weevils remained alive and mobile for at least 24 h. Each pair of sentinel weevils was subjected to one recording session (described in section 2.3), which began at 16:00, when the sentinel preys were installed, and ended after 24 h, at which time the remaining weevils were counted and recovered.

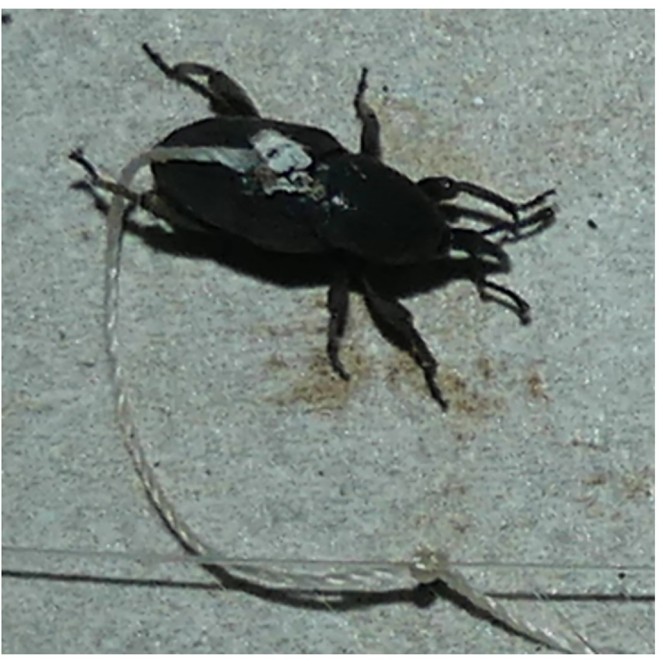

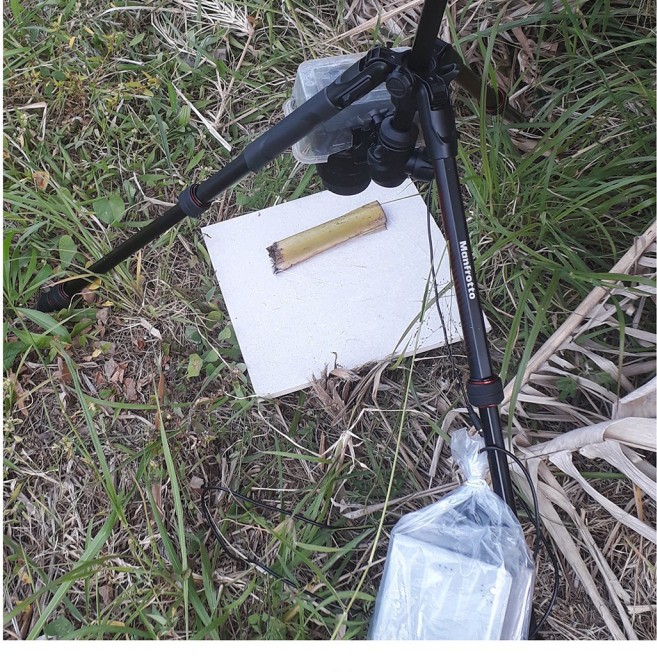

(a)                                                                      (b)

**Fig 1. Sentinel tile with a) an attached *Cosmopolites sordidus* weevil and b) a camera on a tripod.** Because weevils were about 1 cm long, the cropped detail in a) demonstrates the resolution of our images.

## 2.3. Camera setup

We used five Panasonic Lumix DMC-FZ 3000 cameras; each camera was set on a tripod, located above a tile, and facing the tile (see Fig 1b). The tiles on which the weevils were attached provided a homogeneous background. Cameras were powered with external batteries (DLH DY-BE2063, 24 000 mAh), and cameras and batteries were protected from rain by plastic bags. The height of the camera was adjusted such that the width of the image matched the width of the tile. Produced images had a resolution of $3000 \times 4000$ pixels, displaying a surface of $22.5 \times 30.0$ cm with a resolution of 133 px/cm. This high resolution enabled the visual identification of insects and other small animals (see Fig 1b for an example). Focus, aperture, ISO, and flash were automatic. Images were captured every 30 s during the 24-h recording period, which generated about 3000 images per tile per recording period. Given that we had five tiles on each of five plots and five recording sessions per tile, we generated 125 24-hour recordings, which included a total of 312,024 images and representing 3,000 h of observation (600 h per plot).

## 2.4. Image and data analysis

**2.4.1. Predator detection.** The detection of animals passing before a camera may be achieved using sensors (like Passive Infrared sensors, PIR) for the study of medium to large species [45]. However, the use of such technologies for the detection of very small animals presenting no heat differences with the background such as insects is still in development and only possible in controlled conditions (*e.g.* [46, 47]). Detection of passing insects is to our knowledge only achieved via image analysis on image sequences (*e.g.*, [48–50]). As previously suspected predators of *C. sordidus* were mostly insects, we relied on background subtraction to be able to detect both small and larger species. Images were first resized to $300 \times 400$ pixels to ease computations and to discard background noise. In predation events, the predator appears only during a limited number of images, allowing its detection with background subtraction. During a recording session, a rolling average image is computed for every 50 images to provide a background model. All $I_i$ images are then compared to the rolling average image $\hat{I}$ centred on $I_i$ (see Fig 2). The comparison of $I_i$ and $\hat{I}$ is done pixel by pixel. For each pixel $\hat{p}$ and $p_i$ (belonging to $\hat{I}$ and $I_i$ respectively), if $|\hat{p} - p_i| > 70$ on a grey scale (i.e., 256 bits), they are considered different. The source code for image analysis can be found in the following repository: https://gitlab.com/ptresson/rolling_average_background_detection.

To select images presenting sudden variations of content, we calculated the rolling average $\mu$ and the standard deviation $\sigma$ of the proportion of different pixels in 50 images. We then selected images presenting more than $\mu + 2\sigma$ different pixels. These images were then manually reviewed. The selected threshold was chosen using five sessions where predation had occurred for calibration.

**2.4.2. Predation analysis.** After the selection of images displaying animal crossings, the images were reviewed to identify those where predation occurred. Predators were identified in original high-definition images and with a guide of local fauna [51]. An individual observed feeding directly on a weevil or within four frames (2 min) of the disappearance of a weevil was considered a predator. If the predation event was not detected, the session was manually reviewed to identify the frame in which the weevil disappeared.

Every weevil predation was counted as a single predation event, even when both weevils disappeared within the same frame (indicating that both predation events occurred in < 30 s). If a predator did not appear on the images but the weevils disappeared, the predator was labelled "unseen". We also estimated predation frequency as the ratio between the appearances of a predator species during a predation event and their total appearances. All consecutive frames of appearance of an individual predator were counted as a single appearance.

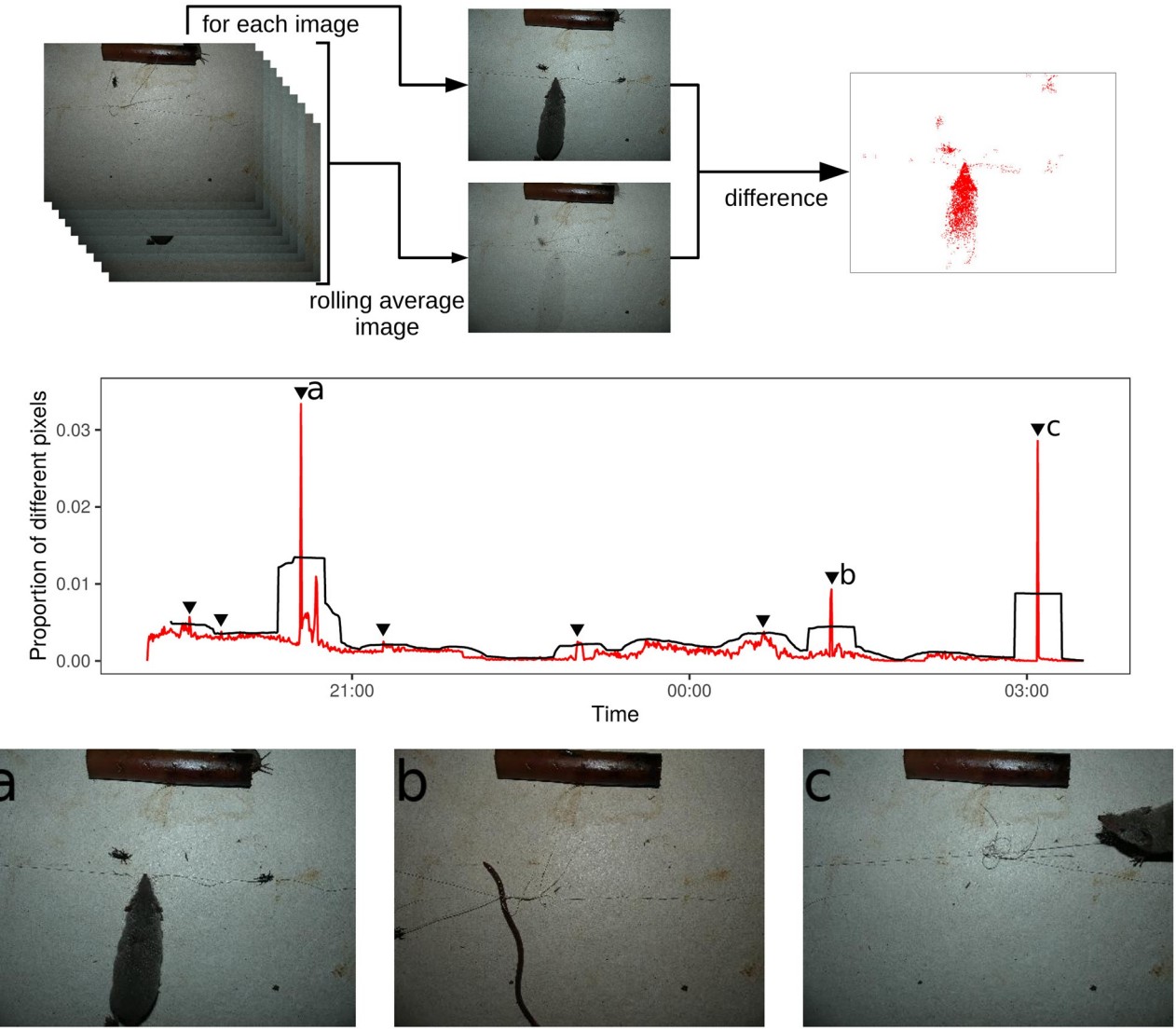

**Fig 2. Example of predator detection with a 1000-image sequence in our dataset.** The black line represents the $\mu + 2\sigma$ threshold for the selection of images. When the proportion of different pixels (red line) exceeds this threshold, images are selected for review (pointers on the graph). Here, 8 of 1000 images are selected, including two that showed predation of *C. sordidus* (*a* and *c*). Other large changes in the image content correspond to detection of other animals, such as an earthworm (*b*).

For each determined predator species, we used a GLM with poisson distribution to test the effect of the hour of the day on the predation events.

## 3. Results

### 3.1. Predator detection

The use of background subtraction enabled us to detect tile crossings by multiple animals. Crossing invertebrates were detected in every session. More than 1800 individuals belonging to 60 species, including ants, spiders, slugs, cockroaches, beetles and earwigs have been detected. In addition, 150 vertebrate individuals have been detected crossing the tiles. The predator was detected and identified in 64 (78.0%) of the 82 total predation events. In the 18

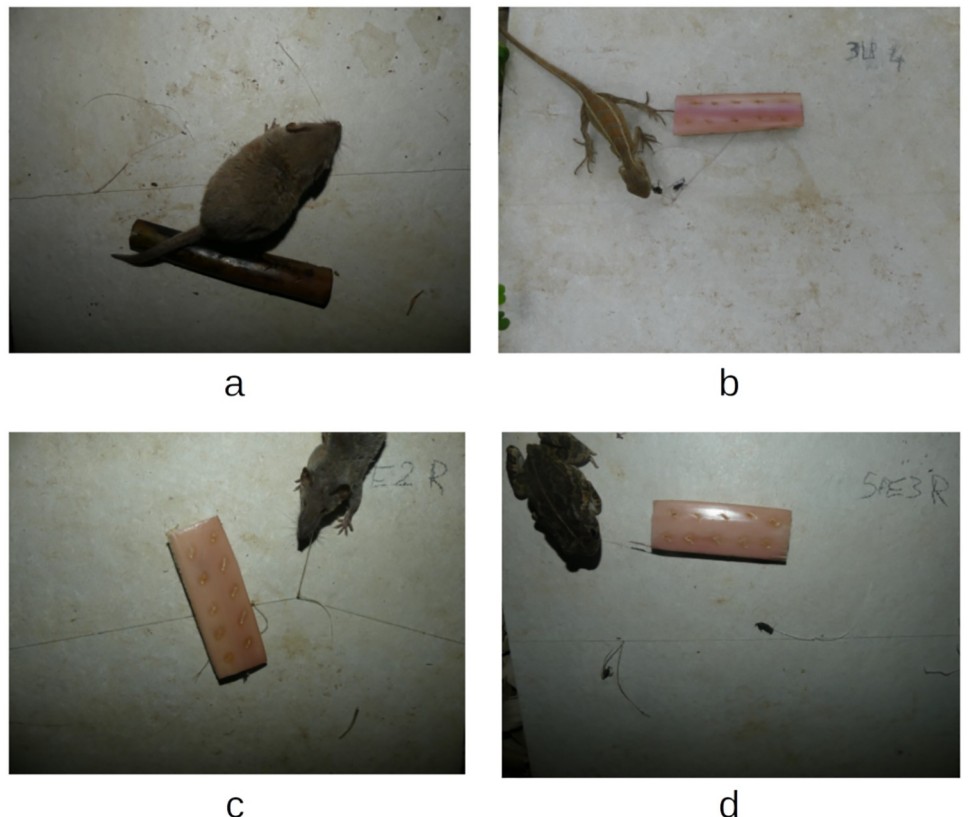

**Fig 3. Images featuring the following predators of *C. sordidus*: a) *Mus musculus*; b) *Calotes versicolor*; c) *Suncus murinus*; and d) *Sclerophrys gutturalis*.**

cases where predators were unseen; 6 predations happened between two frames; 9 happened outside the tile; 1 was obscured by a banana leaf displaced by the wind and 2 because of poor image quality (flash failure). In all cases, clues strongly suggested that the disappearance of the weevils was due to predation such as a strong string tension visible on the frame of disappearance or the appearance of weevil remains on the next frame.

In total, 40 individuals performing predation were observed, with 23 being responsible for the nearly simultaneous predation of the two attached weevils. All recorded predation events involved vertebrates. The identified predators included 34 Asian house shrews (*Suncus murinus*), 3 house mice (*Mus musculus*), 2 oriental garden lizards (*Calotes versicolor*), and one guttural toad (*Sclerophrys gutturalis*) (Fig 3). Although they were detected on the tiles, arthropods were not detected in any predation event. Because of this, we chose to focus only on observed predator species, and data concerning arthropods is therefore not further developed.

### 3.2. Time of predation

The use of images enabled us to determine the exact time of predation during the 24-hour period (Fig 2). All predators except *C. versicolor* were most active at night (Fig 4). All predations by *S. murinus*, *M. musculus*, and *S. gutturalis* occurred between 18:00 and 5:00, which corresponded to night on La Réunion Island. *Suncus murinus* appeared to be mostly active at the beginning of the night. The unseen predation events occurred during day and night (10 events at night and 8 during the day). Of the 82 predated weevils, 43.9, 62.1, and 85.4% have

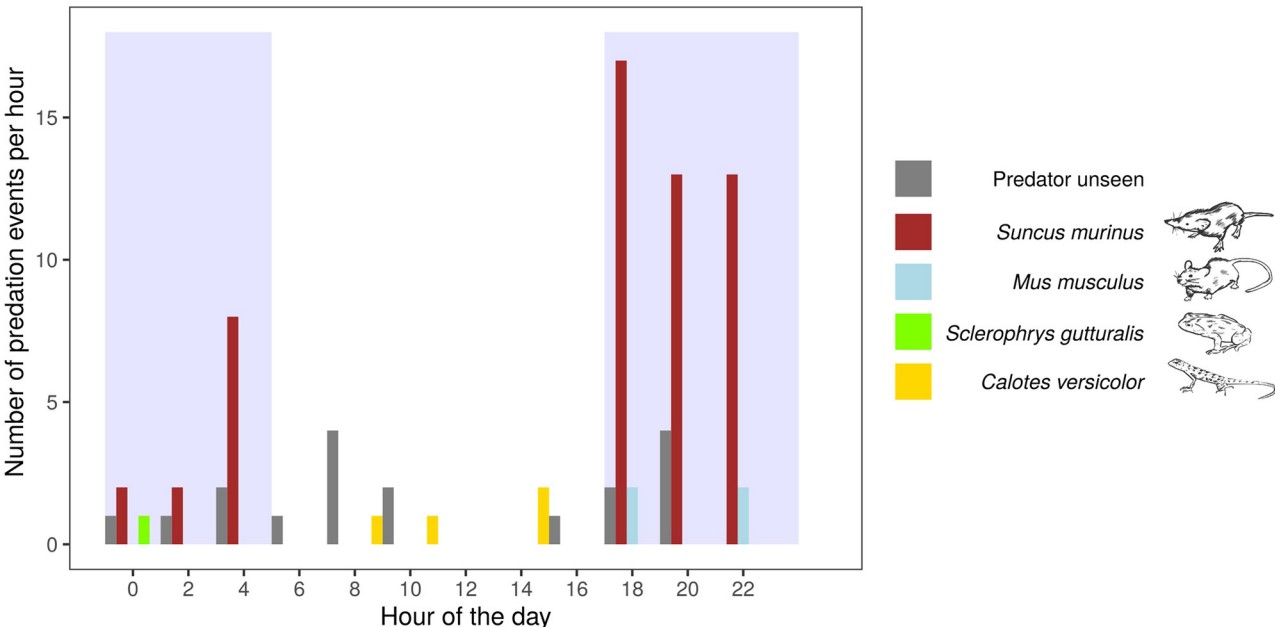

**Fig 4. Number of predation events per hour for different predator species.** The shaded area represents night time.

been consumed within the first 6, 8, and 12 h of prey deployment, respectively. Predation events were significantly affected by the hour of the day for *S. murinus* and *M. musculus* (p-value < 0,001 and 0,004, respectively). For other species the relation was not significant.

### 3.3. Proportion of predation events per identified predator

Among the 82 predation events (which involved 32.8% of the deployed weevils), 55 (67.0%) were done by *S. murinus*, 4 (4.9%) by *M. musculus*, 4 (4.9%) by *C. versicolor*, 1 (1.2%) by *S. gutturalis*, and 18 (21.9%) by unseen predators. *Suncus murinus* achieved predation in all plots (Fig 5). For PE, PC, and LP plots, the predation was achieved by three predator species; in BM and SL plots, the predation was achieved by one or two species, respectively. The predation rate was highest in the PE and LP plots and was lowest in the PC and BM plots (Fig 5). Among the 125 recording sessions, no weevil was predated in 74 and both weevils were predated in 31.

### 3.4. Percentage of predator appearances resulting in predation (predation frequency)

Predation frequency was highest for *C. versicolor*, lowest for *S. gutturalis*, and intermediate for *S. murinus* and *M. musculus* (Fig 6). The most frequently observed predator, *S. murinus*, was associated with predation events about half the time (Fig 6).

## 4. Discussion

### 4.1. Ecological contribution

We found that all the predated weevils were consumed by vertebrates, although reports of predation of *C. sordidus* by vertebrates had previously been considered anecdotal [39, 41]. For the first time, we provide the evidence that rodents (*S. murinus* and *M. musculus*) may be key actors in the regulation of *C. sordidus* given that they were responsible for 71.9% of the recorded predation events. Although lizards and amphibians seemed to be responsible for less

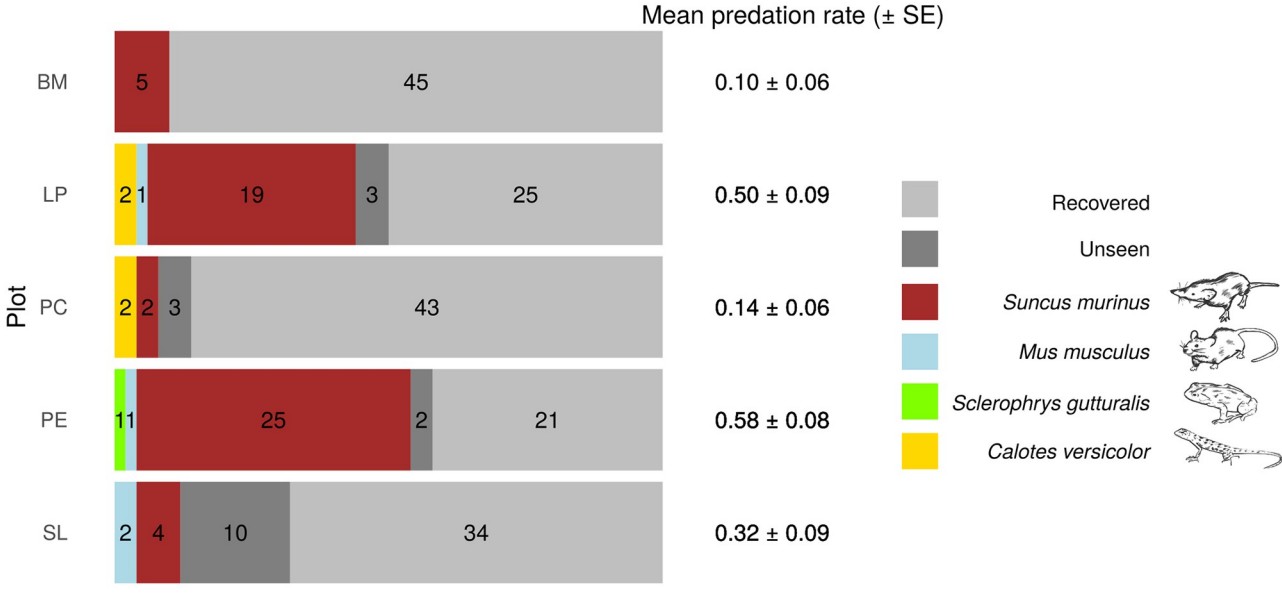

**Fig 5. Total number of predation events per plot and per predator specie and the associated mean predation rate (number of predation events/ number of prey deployed).** SE is calculated considering each sentinel prey replicate on one plot (*i.e. 25 replicates*).

regulation than rodents, their identification as predators of *C. sordidus* extends the range of possible management solutions for controlling this pest in a sustainable manner. Because the observed species of vertebrate predators are widespread in tropical areas, our results may be relevant to other banana-producing regions.

Data on the time of predation during the 24-hour period is useful because it helps us understand the potential effects of the predator on its prey. For instance, the nocturnal activity and foraging habits of *S. murinus* [52] fit the nocturnal activity of the weevil [40] and are therefore well suited for predation of weevils under natural conditions. On the other hand, our assessment of the role of *C. versicolor* in the regulation of *C. sordidus* is likely overestimated, because *C. versicolor* is most active during the day and may not actively forage in search of prey [53]. In other words, the lizard's activity period and behaviour do not match with the nocturnal and hidden behaviour of *C. sordidus*. Information on the time of predation may also allow researchers to hypothesize about the identities of unseen predators.

In the studies reviewed by Lovei and Ferrante [9], the median predation rate on live prey in sentinel prey experiments was 26%, which is consistent with our results. Working in cereal fields in Sweden, Tschumi et al. [29] found that 38.0% of sentinel preys were consumed by predators, 84.5% of which were rodents. Despite the differences in the studied ecosystems (cereal fields vs. banana fields), our results were very similar to those of Tschumi et al. [29]. Indeed, we found that 32.8% of the sentinel weevils were consumed by predators, 71.9% of which were rodents. In the current study in banana fields, *S. murinus* appeared to be the most important predator of *C. sordidus*. We expect the measured contribution of *S. gutturalis* in the regulation of *C. sordidus* to be substantially underestimated because the weather during the study was exceptionally dry. It was the eighth driest November recorded on La Réunion since 1980, with 50% less precipitation compared to the 1981–2010 average. The location of the PC and PE plots (Pierrefonds) received only 2.0 mm of rain during this month [54]. Consequently, ponds in PC and PE were low, leading to reduced activity of frogs and toads.

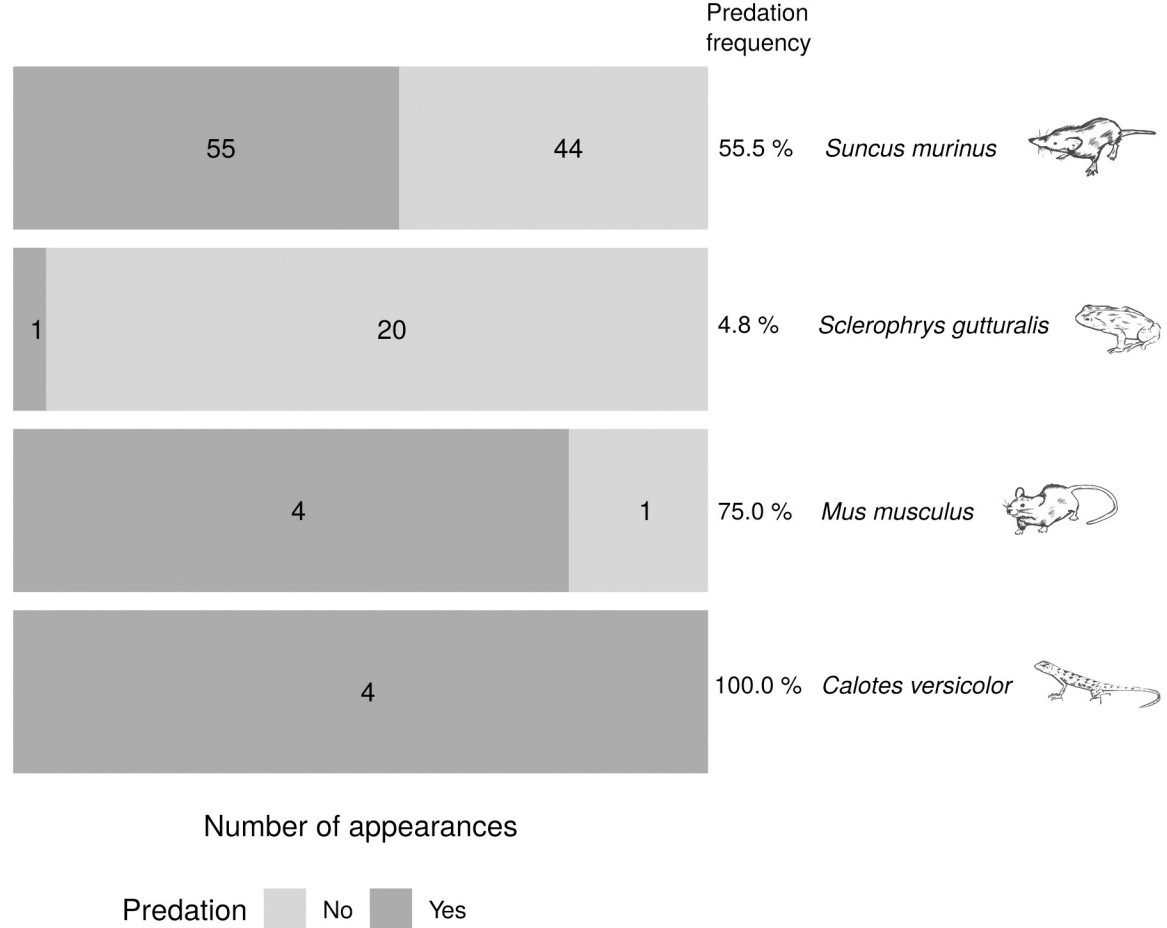

**Fig 6. Number of appearances of the different predators with or without predation and the associated predation frequency.**

We found vertebrates rather than invertebrates to be predators of adult weevils, and previous research indicated that invertebrates (such as ants or earwigs) are likely to be predators of the egg and larval stages of the weevil [41]. However, it has been hypothesized that eggs only remain accessible for predation during a relatively short time because they are rapidly covered with leaking latex after oviposition [39]. Immature stages of *C. sordidus* are hidden in the corm and are thus of difficult access for predators. Furthermore, given longevity and fecundity of *C. sordidus*, modelling studies focused on this pest have suggested that predation of adults rather than immature stages is more likely to affect weevil population dynamics [55, 56]. The ability to distinguish adult, egg, and larval predation could be useful for understanding the regulation of pests. Indeed, predators targeting different life stages may have complementary roles and lead to a better regulation overall [30, 57].

### 4.2. Methodological contribution

The use of images as a source of information made it possible to detect the previously unconfirmed role of vertebrates in the regulation of *C. sordidus*. This confirms the potential utility of images in the study of biodiversity and interactions. By using images, valuable information such as the identification of the predators and the time of predation is easily accessed. In our case, the analysis of 312,024 images required one- half of a workday for one person. The use of

background subtraction rather than deep learning enabled us to obtain useful results without an extensive training dataset, labelling, or powerful computing resources. In addition to easing data analysis, our method was easy to implement and could be applied to any research question in ecology that could benefit from visual information.

Some image sequences in our dataset suggested situations in which DNA metabarcoding could provide biased information concerning trophic interactions. Some sequences, for example, showed the remains of a previously predated weevil that attracted ants that then fed on the remains without being the primary predators. Similarly, another sequence of images (see Fig 7) showed a cockroach scavenging on the remains of a weevil that had been killed and partly consumed by a shrew. In these cases, DNA metabarcoding would have identified the ants and cockroach as predators even though they were only scavengers.

A common limitation of sentinel prey experiments is the inability to identify the predator. As mentioned earlier, in 70% of the studies reviewed by Lovei and Ferrante [9], the identity of

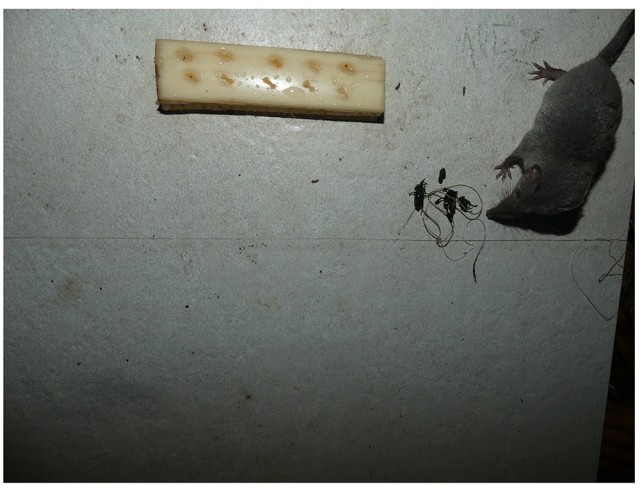

(a)

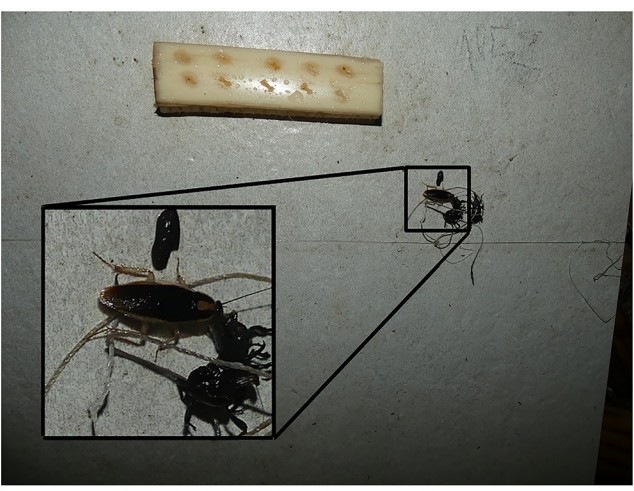

(b)

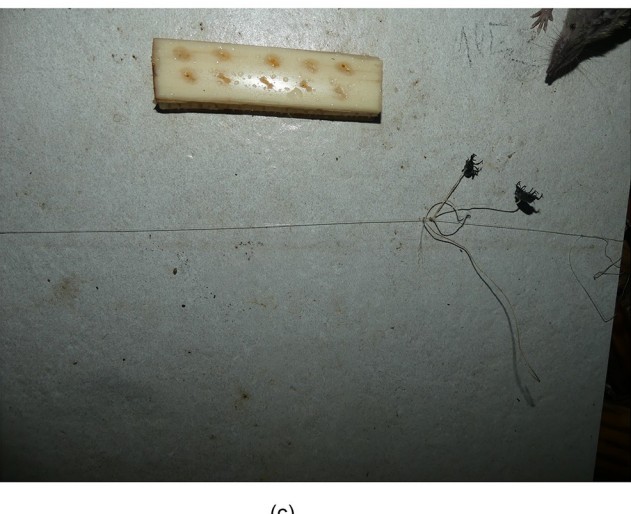

(c)

**Fig 7. Example of hyper-predation.** a) A shrew (*Suncus murinus*) captures a weevil at 19:27. b) A cockroach then feeds on the weevil remains for almost 1 h before c) it is captured by *S. murinus* at 20:24.

predators involved in sentinel prey consumption remained unknown. When the experiment is not monitored (by image capture, for example), vertebrate predators are generally identified via exclusion (*e.g.*, [34, 58–60]), sentinel models (*e.g.*, [61]), or correlation. Potential predators are thus targeted in the experimental design based on size, ability to fly, habitat, or other characteristics. Visual monitoring of sentinel prey experiments reduces these possible biases and allows the detection of predator species that were previously unknown. Only recently have rodents (mice and rats) been identified as major actors in mealworm regulation in Sweden, and this resulted from the monitoring of sentinel prey experiments [29]. Similarly, experiments in China have only recently identified toads as biological control agents of the brown planthopper via monitored sentinel prey experiments [27, 28].

## 4.3. Limitations

No method alone can provide a complete understanding of the functioning of an ecosystem. Although the proposed method opens new perspectives on the predation of this pest, it only captures physical interactions on the ground. Our method would be unfit to the study of flying species (such as bats, see [59]) or predation happening below ground (or in the case of *C. sordidus*, predation of larva inside the corm for instance). Conversely, numerous ecological interactions happen without visual clues or physical interaction, particularly for arthropods that may experience the world more through chemical clues. Behaviours without a physical interaction, such as avoidance of a predator based on environmental cues for instance, can not be recorded using our method.

Our study setup (tiles, cameras with flash) may also have perturbated the behaviour of some species. The weevils remained exposed for long periods of time whereas its natural behaviour would have been to hide during the day. For this reason we consider the observed predation by *C. versicolor* to likely be an experimental artefact. The presence of tiles and camera flashes may also have disturbed the behaviour of some species.

We therefore advocate for the complementarity of methods to get the best understanding possible of ecological interaction. In our case, the potential predation of *C. sordidus* by arthropods had already been studied with other methods [39] and our study brings a new understanding of previous knowledge but would not be sufficient as a first step to study a novel pest in an ecosystem.

## 4.4. Implications for management

Research has indicated that conservation biological control is enhanced by the preservation of natural habitat [62, 63], the diversification of plant species [64], and the maintenance of complex landscapes [62, 65]. Several studies have suggested that the diversification of plant species in banana plots favours the regulation of C. sordidus; such diversification has involved grass cover [12, 66] or an associated crop [67]. The identification of new predators for C. sordidus extends the range of possible management strategies for the control of this pest. In light of our results, we would also recommend that the banana producers provide habitats, such as hedges or grasslands [52], for S. murinus in their plots. On the other hand, S. murinus and M. musculus are considered invasive species in several tropical ecosystems [68]. Moreover, these species may as well be considered as harmful in banana fields as potential hosts to Leptospira [69]. Actions that favour these species must therefore be carefully considered. Because we suspect that the contribution of S. gutturalis to pest regulation was higher than measured in our study, we also recommend management policies to favour the activity of toads and frogs, such as the installation of ponds near or within banana growing plots.

The spatial organisation of plots designed to favour predation (e.g., patches of bare soil and grass cover) warrants additional research. Indeed, some studies suggest that pest control may be influenced at a local scale. For instance, sward heterogeneity in vineyards may affect prey-predator interactions and bird foraging behaviour [32].

Our study in itself does not confirm the relationship between habitat composition and weevil regulation, and further research is needed to evaluate the effects of possible changes in habitat elements on predator abundance and predation rate. Our study also suggests a possible relationship between weevil infestation level and cultural practices on predation of weevils by vertebrates (S1 Fig, Table 1). Before generalizing our results to other regions of the world, il would be needed to carry out other similar experiments in other banana producing regions, including some treatments with additional ponds to favour amphibian species. The methods described in the current report can be used to easily investigate potential predators in other banana growing regions and to clarify the relationships between landscape, habitats, biodiversity, and pest regulation. Knowledge of the key predator species and how to favour them will facilitate the application of conservation biological control.

## 5. Conclusion

The monitoring of sentinel prey experiments allowed us to identify predators of the adult of banana weevil in La Réunion island. The identified predators were all vertebrates, such as the Asian shrew (*Suncus murinus*), the house mouse (*Mus musculus)*, the oriental garden lizard (*Calotes versicolor)*, and the guttural toad (*Sclerophrys gutturalis)*. This is the first time that vertebrates are proven to be predators of *C. sordidus*. The exact time of predation events were determined from the images metadata. It was thus possible to identify predator foraging periods that coincided with activity of adult weevils. The identification of these predators opens new perspectives for the control of the banana weevil, such as the preservation of habitat or the installation of ponds. Our study also confirms the usefulness of image analysis to study ecological interactions. More broadly, it questions our understanding of the role of vertebrate in the biological control of pests and the complementarity of biodiversity sampling methods.

## Supporting information

**S1 Fig. Observed predation rate (mean ± SE) compared to the weevil infestation level in the five plots.** The sanitary ranking of plots was based on assessment of weevil infestation level before the experiment, damage reported by the farmer, and age of the plot; the ranking increased as the values of these variables increased. The PC plot was newly planted with in-vitro propagated banana plants and lacked any history of weevil infestation because the plot was not cultivated before, i.e., the absence of weevil damage in the PC plot was probably not caused by regulation by natural enemies. On the contrary, BM plot is planted with in-vitro propagated plants but banana was already cultivated before and the plot had an already installed weevil population.
(TIF)

## Acknowledgments

The authors would like to thank the reviewers for their insightful comments.

## Author Contributions

**Conceptualization:** Paul Tresson, Philippe Tixier, William Puech, Dominique Carval.

**Data curation:** Paul Tresson.

**Investigation:** Paul Tresson, Bernard Abufera, Antoine Wyvekens, Dominique Carval.

**Methodology:** Paul Tresson, Philippe Tixier, William Puech, Dominique Carval.

**Project administration:** Philippe Tixier, William Puech, Dominique Carval.

**Software:** Paul Tresson.

**Supervision:** Philippe Tixier, William Puech, Dominique Carval.

**Visualization:** Paul Tresson.

**Writing – original draft:** Paul Tresson, Philippe Tixier, Dominique Carval.

**Writing – review & editing:** Paul Tresson, Philippe Tixier, William Puech, Dominique Carval.

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
