## [Decision Letter · Decision Letter 0]

24 Jun 2022

PONE-D-22-13662Caught on camera: Field imagery reveals the unexpected importance of vertebrates for biological control of the banana weevil

(Cosmopolites sordidus Col. Curculionidae)PLOS ONE

Dear Dr. Tresson,

Thank you for submitting your manuscript to PLOS ONE. After careful consideration, we feel that it has merit but does not fully meet PLOS ONE’s publication criteria as it currently stands. Therefore, we invite you to submit a revised version of the manuscript that addresses the points raised during the review process.

We look forward to receiving your revised manuscript.

Kind regards,

Bi-Song Yue, Ph.D

Academic Editor

PLOS ONE

Journal Requirements:

Reviewers' comments:

Reviewer's Responses to Questions

**Comments to the Author**

1. Is the manuscript technically sound, and do the data support the conclusions?

Reviewer #1: Yes

Reviewer #2: Partly

2. Has the statistical analysis been performed appropriately and rigorously? 

Reviewer #1: N/A

Reviewer #2: Yes

3. Have the authors made all data underlying the findings in their manuscript fully available?

Reviewer #1: Yes

Reviewer #2: Yes

4. Is the manuscript presented in an intelligible fashion and written in standard English?

Reviewer #1: Yes

Reviewer #2: Yes

5. Review Comments to the Author

Reviewer #1: I have finished reading this interesting paper where authors registered several vertebrate predator species preying on C. sordidus using field imagery. In my opinion, they present some relevant and novel data. Furthermore, the use of the images provides indisputable data on the role of predators that the authors have recorded. Despite these merits, my main concerns about this study are:

The introduction should be better presented regarding the terms and the knowledge gap addressed by the paper.

There is no inferential statistical analysis to support the data presented.

The discussion does not present the limitations of the method and its advantages for studies of this nature.

I hope my comments and suggestions can help authors improve their interesting manuscript.

ABSTRACT

- L26-27: As I will argue below, I believe that saying that "the role of vertebrates on biological control may be underestimated" suits better here.

- L 43: In my opinion, your study could be extrapolated to other pest and vertebrate groups. Therefore, I suggest adding that "… of the banana weevil, possibly also favoring other vertebrate and invertebrate natural enemies." Please, see further comments on this topic below.

- Keywords: I suggest adding the keyword "ecosystem services" due to the relevance of this study to the subject.

INTRODUCTION

- L50: This reference is interesting, but it is outdated. I believe that the study from Perovic et al. 2017. Biological Reviews 93: 306-321, and the references therein could be helpful to the authors.

- L50-61: Here, I believe that there are some flaws in some relevant concepts and terms used by the authors. First, I believe that the authors should state at the beginning of the paragraph that biological control is an ecosystem service provided by biodiversity, where natural enemies regulate their prey population densities. Second, conservation biological control is one of the biological control strategies available. As stated in some references used by the authors (e.g., Shields et al. 2019), conservation biological control strategies rely on habitat management to favor the occurrence and functionality of naturally occurring natural enemies in a given site. That could be done by several means, including plot and landscape-scale management. Therefore, it is not the control itself that matters apart from the management strategy. Third, the assumption on L54-55 is not entirely true since biological control could be provided by keystone or by multiple species simultaneously (complementary effects of biodiversity). That is because CBC strategies rely more often on generalist rather than specialist species where multiple species (such as vertebrate and invertebrate species) play a role (see reference 12 you cited). Therefore, I suggest the author focuses on the relevance of knowing the species available to provide biological control to manipulate further or favor community-level interactions.

- Another flaw I believe that authors should better situate the discussion about the limitations of sampling methods. All sampling methods have limitations, including the approach used by the authors in this paper. For example, a recently published paper (Aguiar et al. 2021. Plos One 16(10): e0258066) on the consumption of agriculture pests by bats in urban areas used the DNA metabarcoding approach to identify pests predated by bat species. Your approach would not be helpful to this study. However, your approach is very useful for other cases and other organisms. Therefore, I suggest authors focus on how different methods can complement each other depending on the study system. Specifically, I believe that this suggestion could help rearrange L62 to L80.

- L94-96: I did not get the point here. If a study was designed to evaluate the role of arthropods in pest control, then vertebrates would be almost automatically neglected. Don't you think that the main gap here would be further investigating the role of vertebrates in biological control using specific methods for this, as you have done? Please, clarify this topic here. I also suggest reading Aguiar et al. 2021. Plus One 16(10): e0258066.

- L109: Is the biological control provided by insects really ineffective? I read the paper you cited, and there the authors say that ants provide biological of this pest species. I agree that C. sordidus adults have fewer invertebrate predators than other pest insects, but the central gap here, in my opinion, is that vertebrates could complement the role of invertebrate species by attacking different stages of development of the pest. That is very interesting for CBC strategies and increases the importance of biodiversity in providing ecosystem services, such as biological control.

- L115: Something is missing here.

M&M

- L131-132: I suggest adding some biological information about C. sordidus (e.g., they are usually aggregated in the field and are usually hidden inside the banana stems) to justify the independency of some plots.

- L 142-148: It is not clear to me how many samples you did for each sample site. I would ask the authors to be more specific on how many samples through time were made. Could you please specify these points? How were the sentinel preys obtained? Did you rear them or collect them in the field?

- L162-175 and 181-204: When I saw the sample pictures, I was in doubt whether the cameras would be able to identify small arthropod predation. The equipment height seems too high to detect small arthropod predation in front of the camera using the approach you described in L 187-188 and the thresholds for selecting images in Figure 2. Could you please clarify how you discarded that a passing recorded arthropod in front of the camera was predating or not the sentinel prey?

- I missed some inferential statistics to compare predation data through time and predation rate among species for all the data presented by the authors. I suggest adding some simple statistics corrected to small samples to increase the robustness of this study and to confirm the patterns observed by the authors. In my opinion, this is a critical point to be considered.

DISCUSSION

- The discussion is interesting, but some points should be rearranged. My primary concern is that the authors did not recognize the limitations of their methods. First, a white paper under the sentinel prey might have deterred some arthropods from assessing their prey. Many arthropods with poor vision use other environmental cues such as plant volatiles to find their prey. Such stimulus was not presented to potential predators when you use sentinel prey. The sentinel prey was also more exposed to predation than in natural conditions because this species is usually sheltered inside the banana plant stems. Also, the observation time (24 hours) may not have been sufficient to capture other predator species, especially invertebrates. These factors do not demerit the study in any way. However, they reinforce that it is necessary to use a combination of methods to understand the complex ecological interactions in agroecosystems and to understand the role of different species in biological control. I suggest that the authors add a section on these limitations as a basis for other more complex studies on the complementarity of biodiversity in pest control in the future. I also suggest focusing on how the approach used by the authors can be complementary to other methods and reveal unexpected interactions that other methods failed to capture.

- I am not sure if vertebrates are key actors (L295) in regulating C. sordidus populations because it may depend on the methods used in field trials and the species pool available for predating the insect. In other regions, such as the tropics, arthropods and other vertebrate species may be more relevant. For example, a farm without a pound nearby may not have amphibians as part of the predatory species pool. Therefore, I suggest authors restrict these extrapolations to their study region and suggest that more studies with vertebrates and invertebrates should be done in other regions to confirm their results. However, I suggest expanding the discussion on complementary effects (L334-337) and the relevance of species with distinct traits in providing biological control. That will provide more robust evidence to farmers and decision-makers that biodiversity matters for pest control and for designing biodiversity-friendly landscapes. Nevertheless, as the authors mentioned (L396), their study was not designed to account for evaluating the role of habitat management in biological control. I suggest therefore reducing this discussion to open more space to discuss the methodological limitations of their study, as I mentioned before. In my opinion, this discussion would fit better in a conclusion section.

- I suggest authors add a conclusion section to their paper that answers the main questions stated in the introduction and indicates the new steps needed to understand the role of vertebrate predators in conservation biological control more broadly.

Reviewer #2: The manuscript appears technically sound and the overall design of the project is very well thought out and properly executed. The methodology is explained clearly and succinctly with enough detail to allow for the experiments to be independently replicated. Sufficient justification is given in the text to justify both the novelty of the project as well as the specific methodologies chosen. No specific hypotheses are given however three main research questions are clearly and succinctly outlined in the introduction. Overall, I do not see the lack of specific hypotheses as problematic as the authors clearly intend the project as more of an exploratory study as opposed to a series of specific manipulative experiments.

There is limited scope for statistical analyses in this publication given the relatively simplistic nature of the data and research questions and the lack of a specific hypothesis as outlined in the introduction. Nonetheless, in the few instances where statistics are calculated (for example means) they are appropriately reported with standard errors.

Generally, the language used in the manuscript is intelligible and of acceptable quality, however I have suggested a number of minor corrections to grammar or phrasing which can be found in the attached document.

The ecological and methodological findings of the study both appear novel and useful for further research in this area. The collection of specific evidence of vertebrate predation of C. sordidus confirms previously reported anecdotal evidence and is crucial for development of future biological control programmes for this pest. Similarly, the use of sentinel traps paired with image background subtraction provides proof of concept for an efficient system of monitoring predation of pest insects while retaining important information allowing for the accurate identification of predator species and eliminating possible biases or confounding factors that may be present when using other methods.

Finally, while the conclusions around the utility of this research methodology in future studies and the conclusions that future studies need to focus more on vertebrate predators of C. sordidus are well supported by the results, the other implications for management included in section 4.3 are more problematic.

The authors are correct to be cautious about implying any causal relationship between weevil infestation level and cultural practices, as this is not tested thoroughly through this study. However I would note that similarly, their recommendations regarding provision of habitats for S. murinus, M. musculus and S. gutturalis are also only very weakly supported by the data in the study and would require further, hypothesis-driven manipulative experiments to be properly justified. These recommendations should also take into consideration whether these species are themselves considered pests of banana or plantain production via possible consumption of banana plants, as well as the potential for these predators to cause other potential issues for farmers, (for example, acting as potential hosts of infectious human pathogens such as Leptospira).

In general, I think the authors need to be careful about making these recommendations based purely on this study alone and instead stick to the main recommendations that can be supported by the data: that future studies on C. sordidus biological control need to focus on vertebrate predators, in particular the role of small rodents.

In conclusion, I think this is a sound paper with exciting and novel results which should be easily accepted after only minor revisions outlined in the attached document and some rethinking of the recommendations in the final section.

6. PLOS authors have the option to publish the peer review history of their article (what does this mean?). If published, this will include your full peer review and any attached files.

Reviewer #1: No

Reviewer #2: **Yes: **Callum John Edwin Thomas

---

## [Author Response · Author response to Decision Letter 0]

7 Jul 2022

Editor

The manuscript as been formatted according to this template.

Not applicable to our dataset because we made all acquisitions ourselfs.

The grant numbers should match now.

 Our dataset is on dataverse.cirad.fr, accessible at this url :

https://dataverse.cirad.fr/privateurl.xhtml?token=a0c04140-35ed-48ec-9e8b-5f113a7c7d94

The caption has been included as needed.

Reviewer 1:

ABSTRACT

- L26-27: As I will argue below, I believe that saying that "the role of vertebrates on biological control may be underestimated" suits better here.

This sentence has been changed accordingly L27-28.

- L 43: In my opinion, your study could be extrapolated to other pest and vertebrate groups. Therefore, I suggest adding that "… of the banana weevil, possibly also favoring other vertebrate and invertebrate natural enemies." Please, see further comments on this topic below.

This sentence has been added as suggested L45.

- Keywords: I suggest adding the keyword "ecosystem services" due to the relevance of this study to the subject.

This keyword has been added as suggested.

INTRODUCTION

- L50: This reference is interesting, but it is outdated. I believe that the study from Perovic et al. 2017. Biological Reviews 93: 306-321, and the references therein could be helpful to the authors.

This reference has been updated as suggested L52.

- L50-61: Here, I believe that there are some flaws in some relevant concepts and terms used by the authors. First, I believe that the authors should state at the beginning of the paragraph that biological control is an ecosystem service provided by biodiversity, where natural enemies regulate their prey population densities. Second, conservation biological control is one of the biological control strategies available. As stated in some references used by the authors (e.g., Shields et al. 2019), conservation biological control strategies rely on habitat management to favor the occurrence and functionality of naturally occurring natural enemies in a given site. That could be done by several means, including plot and landscape-scale management. Therefore, it is not the control itself that matters apart from the management strategy. Third, the assumption on L54-55 is not entirely true since biological control could be provided by keystone or by multiple species simultaneously (complementary effects of biodiversity). That is because CBC strategies rely more often on generalist rather than specialist species where multiple species (such as vertebrate and invertebrate species) play a role (see reference 12 you cited). Therefore, I suggest the author focuses on the relevance of knowing the species available to provide biological control to manipulate further or favor community-level interactions.

This paragraph has been reshaped as suggested, see L55-61.

We now mention the notion of service and of species assemblages.

- Another flaw I believe that authors should better situate the discussion about the limitations of sampling methods. All sampling methods have limitations, including the approach used by the authors in this paper. For example, a recently published paper (Aguiar et al. 2021. Plos One 16(10): e0258066) on the consumption of agriculture pests by bats in urban areas used the DNA metabarcoding approach to identify pests predated by bat species. Your approach would not be helpful to this study. However, your approach is very useful for other cases and other organisms. Therefore, I suggest authors focus on how different methods can complement each other depending on the study system. Specifically, I believe that this suggestion could help rearrange L62 to L80.

A section has been added in the discussion on the limitations of our method and its complementarity with other methods, see section 4.3.

- L94-96: I did not get the point here. If a study was designed to evaluate the role of arthropods in pest control, then vertebrates would be almost automatically neglected. Don't you think that the main gap here would be further investigating the role of vertebrates in biological control using specific methods for this, as you have done? Please, clarify this topic here. I also suggest reading Aguiar et al. 2021. Plus One 16(10): e0258066.

This sentence was reformulated to be clearer, L103-106.

- L109: Is the biological control provided by insects really ineffective? I read the paper you cited, and there the authors say that ants provide biological of this pest species. I agree that C. sordidus adults have fewer invertebrate predators than other pest insects, but the central gap here, in my opinion, is that vertebrates could complement the role of invertebrate species by attacking different stages of development of the pest. That is very interesting for CBC strategies and increases the importance of biodiversity in providing ecosystem services, such as biological control.

This section was reformulated L118-121.

- L115: Something is missing here.

Sentence was completed L128.

M&M

- L131-132: I suggest adding some biological information about C. sordidus (e.g., they are usually aggregated in the field and are usually hidden inside the banana stems) to justify the independency of some plots.

Additional information has been added as suggested, L144-146.

- L 142-148: It is not clear to me how many samples you did for each sample site. I would ask the authors to be more specific on how many samples through time were made. Could you please specify these points? How were the sentinel preys obtained? Did you rear them or collect them in the field?

This has been clarified L164

Precisions on sentinel preys have been added L169-171.

- L162-175 and 181-204: When I saw the sample pictures, I was in doubt whether the cameras would be able to identify small arthropod predation. The equipment height seems too high to detect small arthropod predation in front of the camera using the approach you described in L 187-188 and the thresholds for selecting images in Figure 2. Could you please clarify how you discarded that a passing recorded arthropod in front of the camera was predating or not the sentinel prey?

The identification of arthropods was possible because we used full resolution images, as explained L192. The image presented in figure 2 has been resized for computational efficiency (but suitable for vertebrate determination) as stated in method L214.

- I missed some inferential statistics to compare predation data through time and predation rate among species for all the data presented by the authors. I suggest adding some simple statistics corrected to small samples to increase the robustness of this study and to confirm the patterns observed by the authors. In my opinion, this is a critical point to be considered.

We conducted the suggested analysis, see L253-254 in Methods and L291-283 in Results.

DISCUSSION

- The discussion is interesting, but some points should be rearranged. My primary concern is that the authors did not recognize the limitations of their methods. First, a white paper under the sentinel prey might have deterred some arthropods from assessing their prey. Many arthropods with poor vision use other environmental cues such as plant volatiles to find their prey. Such stimulus was not presented to potential predators when you use sentinel prey. The sentinel prey was also more exposed to predation than in natural conditions because this species is usually sheltered inside the banana plant stems. Also, the observation time (24 hours) may not have been sufficient to capture other predator species, especially invertebrates. These factors do not demerit the study in any way. However, they reinforce that it is necessary to use a combination of methods to understand the complex ecological interactions in agroecosystems and to understand the role of different species in biological control. I suggest that the authors add a section on these limitations as a basis for other more complex studies on the complementarity of biodiversity in pest control in the future. I also suggest focusing on how the approach used by the authors can be complementary to other methods and reveal unexpected interactions that other methods failed to capture.

A section as been added as suggested, see section 4.3.

- I am not sure if vertebrates are key actors (L295) in regulating C. sordidus populations because it may depend on the methods used in field trials and the species pool available for predating the insect. In other regions, such as the tropics, arthropods and other vertebrate species may be more relevant. For example, a farm without a pound nearby may not have amphibians as part of the predatory species pool. Therefore, I suggest authors restrict these extrapolations to their study region and suggest that more studies with vertebrates and invertebrates should be done in other regions to confirm their results. However, I suggest expanding the discussion on complementary effects (L334-337) and the relevance of species with distinct traits in providing biological control. That will provide more robust evidence to farmers and decision-makers that biodiversity matters for pest control and for designing biodiversity-friendly landscapes. Nevertheless, as the authors mentioned (L396), their study was not designed to account for evaluating the role of habitat management in biological control. I suggest therefore reducing this discussion to open more space to discuss the methodological limitations of their study, as I mentioned before. In my opinion, this discussion would fit better in a conclusion section.

Sentences have been added covering this point L461-468.

Complementary effects and management perspectives are mentioned in conclusion as well.

- I suggest authors add a conclusion section to their paper that answers the main questions stated in the introduction and indicates the new steps needed to understand the role of vertebrate predators in conservation biological control more broadly.

A conclusion section was added as suggested.

Reviewer #2: 

The authors are correct to be cautious about implying any causal relationship between weevil infestation level and cultural practices, as this is not tested thoroughly through this study. However I would note that similarly, their recommendations regarding provision of habitats for S. murinus, M. musculus and S. gutturalis are also only very weakly supported by the data in the study and would require further, hypothesis-driven manipulative experiments to be properly justified. These recommendations should also take into consideration whether these species are themselves considered pests of banana or plantain production via possible consumption of banana plants, as well as the potential for these predators to cause other potential issues for farmers, (for example, acting as potential hosts of infectious human pathogens such as Leptospira).

This paragraph has been rephrased and completed as suggested L445-448.

In general, I think the authors need to be careful about making these recommendations based purely on this study alone and instead stick to the main recommendations that can be supported by the data: that future studies on C. sordidus biological control need to focus on vertebrate predators, in particular the role of small rodents.

A sentence has been added to highlight this point L461-468.

All minor corrections of Reviewer 2 have been taken into account in the revised manuscript.

---

## [Editor Report · Decision Letter 1]

18 Aug 2022

PONE-D-22-13662R1Caught on camera: Field imagery reveals the unexpected importance of vertebrates for biological control of the banana weevil (Cosmopolites sordidus Col. Curculionidae)PLOS ONE

Dear Dr. Tresson,

Thank you for submitting your manuscript to PLOS ONE. After careful consideration, we feel that it has merit but does not fully meet PLOS ONE’s publication criteria as it currently stands. Therefore, we invite you to submit a revised version of the manuscript that addresses the points raised during the review process. I found you pasted your references several times. Please check your manuscript carefully and resubmit again. Please submit your revised manuscript by Oct 02 2022 11:59PM. If you will need more time than this to complete your revisions, please reply to this message or contact the journal office at plosone@plos.org. Please include the following items when submitting your revised manuscript:A rebuttal letter that responds to each point raised by the academic editor and reviewer(s). You should upload this letter as a separate file labeled 'Response to Reviewers'.A marked-up copy of your manuscript that highlights changes made to the original version. You should upload this as a separate file labeled 'Revised Manuscript with Track Changes'.An unmarked version of your revised paper without tracked changes. You should upload this as a separate file labeled 'Manuscript'.If applicable, we recommend that you deposit your laboratory protocols in protocols.io to enhance the reproducibility of your results. Protocols.io assigns your protocol its own identifier (DOI) so that it can be cited independently in the future. For instructions see: https://journals.plos.org/plosone/s/submission-guidelines#loc-laboratory-protocols. Additionally, PLOS ONE offers an option for publishing peer-reviewed Lab Protocol articles, which describe protocols hosted on protocols.io. Read more information on sharing protocols at https://plos.org/protocols?utm_medium=editorial-email&utm_source=authorletters&utm_campaign=protocols.

We look forward to receiving your revised manuscript.

Kind regards,

Bi-Song Yue, Ph.D

Academic Editor

PLOS ONE
---

## [Author Response · Author response to Decision Letter 1]

19 Aug 2022

Comment by academic editor:

I found you pasted your references several times. Please check your manuscript carefully and resubmit again.

Response:

There was indeed issues with our references sections. The references have been checked carefully.

---

## [Editor Report · Decision Letter 2]

24 Aug 2022

Caught on camera: Field imagery reveals the unexpected importance of vertebrates for biological control of the banana weevil (Cosmopolites sordidus Col. Curculionidae)

PONE-D-22-13662R2

Dear Dr. Tresson,

We’re pleased to inform you that your manuscript has been judged scientifically suitable for publication and will be formally accepted for publication once it meets all outstanding technical requirements.

Kind regards,

Bi-Song Yue, Ph.D

Academic Editor

PLOS ONE

---

## [Editor Report · Acceptance letter]

9 Sep 2022

PONE-D-22-13662R2 

Caught on camera: Field imagery reveals the unexpected importance of vertebrates for biological control of the banana weevil (*Cosmopolites sordidus* Col. Curculionidae) 

Dear Dr. Tresson:

I'm pleased to inform you that your manuscript has been deemed suitable for publication in PLOS ONE. Congratulations! Your manuscript is now with our production department. 

Kind regards, 

on behalf of

Dr. Bi-Song Yue 

Academic Editor

PLOS ONE